# Effect of Impact Angle on the Impact Mechanical Properties of Bionic Foamed Silicone Rubber Sandwich Structure

**DOI:** 10.3390/polym15030688

**Published:** 2023-01-29

**Authors:** Di Zhang, Hui Dong, Shouji Zhao, Wu Yan, Zhenqing Wang

**Affiliations:** College of Aerospace and Civil Engineering, Harbin Engineering University, Harbin 150001, China

**Keywords:** red-eared slider turtle, foamed silicone rubber sandwich structure, finite element simulation, impact angle, damage mode, impact resistance

## Abstract

In this paper, a red-eared slider turtle is used as the prototype for the bionic design of the foamed silicone rubber sandwich structure. The effect of impact angle on the performance of the foamed silicone rubber sandwich structure against low-velocity impact is studied by the finite element method. The numerical model uses the intrinsic structure model of foamed silicone rubber with porosity and the three-dimensional Hashin fiberboard damage model. The validity of the model was verified after experimental comparison. Based on the finite element simulation of different impact angles and velocities, the relationship between impact velocity and residual velocity, as well as the penetration threshold at various impact angles are obtained, and the change law of impact resistance of foamed silicone rubber sandwich structure with impact angle and velocity, as well as the damage pattern of sandwich structure at different impact angles and velocities are given. The results can provide a basis for the impact resistance design of the bionic foamed silicone rubber sandwich structure. The results show that, at a certain impact speed, the smaller the impact angle, the longer the path of the falling hammer along the plane of the sandwich structure, the lighter the damage to the sandwich structure and the greater the absorbed energy, so that avoiding the impact from the frontal side of the sandwich structure can effectively reduce the damage of the sandwich structure. When the impact angle is greater than 75°, the difference in impact resistance performance is only 2.9% compared with 90°, and the impact angle has less influence on the impact resistance performance at this time.

## 1. Introduction

In recent years, biostructural materials have gained great interest because they exhibit mechanical properties that far exceed those of synthetic materials [1,2,3,4,5]. As a typical sandwich structure, they have evolved a complex hierarchy through long-term natural selection, usually consisting of a keratin layer, a dorsal cortical layer, a middle foamy cancellous bone layer, and an abdominal cortical layer, which is considered a defense structure against environmental intrusion to resist extreme mechanical forces, which include sharp, high-strain-rate attacks of crocodiles [6,7]. In this paper, we designed a bionic sandwich structure with unique properties based on the red-eared slider turtle shell, which can be used in protective armor and panel structures such as aircraft wings, floors, and ship hull shells.

So far, researchers at home and abroad have carried out some experimental and numerical simulations of composite structural materials of bionic tortoiseshell back armor. Xu Zhang [8] systematically analyzed the damage behavior of tortoise shells under different immersion times and impact cycles and investigated the impact kinetic behavior of tortoise shells during impact wear, which provided a reliable experimental basis for the design of bionic materials. Professor Fengchun Jiang’s group at Harbin Engineering University designed a multilayer composite material with alternating superimposed titanium metal plates and silicon carbide fiber-reinforced aluminum matrix composite plates, and the strength and toughness of the composite structural material were significantly improved [9]. Pei, BQ [10] studied the chemical composition of the shell structure and its mechanical properties by investigating the composition of the compounds in various parts of the shell, based on the shell keratin sheath and spongy bone. Based on the microstructure of the carapace keratin sheath, spongy bone and spine, a bionic sandwich structure consisting of plate, core and back plate was designed by using modeling software. The impact resistance of the bionic structure was verified by finite element analysis and drop hammer experiments. Numerical results showed that all the bionic structures showed different degrees of impact resistance improvement compared to the control group. Prasad, N [11] generated functionally graded two-stage fiber concrete composed of steel and polypropylene fibers based on the impact resistance bionic of turtle shells to enhance impact resistance and damage mitigation.

Considering that the turtle shell may be impacted from all directions along with the purpose of the designed synthetic bionic sandwich structure, these areas are vulnerable to the impact of foreign objects, such as debris thrown from the runway during landing, birds in the sky, etc., which may cause serious damage to the target structure [12]. At the same time, in a real situation, there are very few cases that completely satisfy the vertical impact, the impact of foreign objects on the sandwich structure often comes with a certain angle, and the direct adoption of the findings of positive impact to explain the widely occurring events of different impact angles, in reality, is not based on sufficient evidence. The specific damage pattern of the sandwich structure after the impact of foreign objects is very complex, and the damage and energy absorption become more complicated when coupled with the frictional slip effect existing at different impact angles [13]. In this paper, a survey of the literature on the effect of the impact angle on sandwich structures was conducted. Pascal et al. [14] studied the damage mechanism of sandwich structures with different stacking sequences at tilt angles of less than 15 degrees. Boonkong et al. [15] experimentally investigated the low-velocity impact response of curved aluminum alloy sandwich panels at different impact angles, analyzed their energy absorption characteristics and corresponding failure mechanisms, and found that the perforation increased with the increase of the impact angle. Chen, Kai et al. [16] explored the effect of impact angle on the dynamic response of a steel trapezoidal corrugated sandwich plate in their simulation model and concluded that the reason why the dynamic response of the structure is more affected by the impact angle is the different contact area between the falling hammer and the structure due to the difference in impact angle.

Although a great deal of research has been conducted on tortoiseshell back armor bionics, it has been limited to three-layer sandwich structures composed of foam and honeycomb, balsa wood, etc. Compared with these sandwich structures, there is a lack of more comprehensive experimental studies on rubber sandwich structures. Foamed silicone rubber sheet is one of the good core materials. As a kind of rubber with low density, it is light, soft, elastic, not easy to transfer heat, and has excellent mechanical properties such as anti-shock, impact mitigation, thermal insulation, and sound insulation. In addition, although a lot of research has been conducted on the mechanical properties of sandwich structures under different impact angles, when the impact angle is large, the test conditions are limited by the test and the test is difficult to implement, and scholarly studies of the oblique impact conditions are often based on a specific angle; the mechanical properties of rubber sandwich structures under different impact angles are still lacking a more comprehensive study. Under impact conditions, the sandwich structure will definitely produce damage, which is also inevitably accompanied by the transformation and transfer of energy, and it is important to explore the damage mechanism to improve the impact resistance of the bionic sandwich structure. In view of this, this paper focuses on the analysis and discussion of the impact angle and speed of the impact on the effect of damage on the foamed silicone rubber sandwich structure.

## 2. Materials and Methods

### 2.1. Bionic Sandwich Structure Design

The adult red-eared slider turtle has a long oval body, a gently elevated dorsal carapace with distinct ridges, and a serrated posterior edge, and it is considered to be an invasive species. The microstructural characteristics of the red-eared slider turtle obtained from the scanning electron microscopy (SEM) results in the literature [6] are shown in (b) in Figure 1. As shown in the figure, the armor is composed of a sandwich composite structure that consists of a relatively dense exterior covering the interior of a fiber foam network, a biomimetic design based on its structural features. The sandwich structure in this study is composed of PVB/ethanol solution coating, unidirectional carbon fiber layer, and foamed silicone rubber sandwich layer, taking into account the size of the tortoiseshell back armor structure and the size of the impact test specimen. The size of the sandwich structure adopts 2.5 times the size of the same layer level of the tortoiseshell back armor, and the size of its structure is 100 mm × 100 mm × 8.9 mm. The thickness of each layer of PVB/ethanol solution coating is 5 μm, there are a total of 60 layers, total thickness is 300 μm, thickness of unidirectional carbon fiber layer is 0.15 mm, density is 300 g/m^2^, thickness of foam silicone rubber is 5 mm, and density is 950 kg/m^3^. Figure 1c represents the stacking order of sandwich structure, respectively, coating (Co)_60_/(0°)_6_/(90°)_6_/Core/(0°)_6_/(90°)_6_, the stacking order is top-down, and the subscript indicates the number of repetitions.

### 2.2. Ontological Model and Numerical Simulation Method

#### 2.2.1. Composite Laminate Damage Criterion

The failure criterion is necessary to predict the failure of composite laminates under composite stress conditions. In the past decades, the 3D Hashin failure criterion is the most commonly used criterion in research. Thus, the 3D failure criterion based on the Hashin failure model is described as follows [17,18,19]:

Fiber tensile failure $\left(\sigma_{11}\geq 0\right)$
(1)$${\left(\frac{\sigma_{11}}{X_{T}}\right)}^{2}+{\left(\frac{\sigma_{12}}{S_{12}}\right)}^{2}+{\left(\frac{\sigma_{13}}{S_{13}}\right)}^{2}\geq 1$$

Fiber compression failure $\left(\sigma_{11}<0\right)$
(2)$${\left(\frac{\sigma_{11}}{X_{C}}\right)}^{2}\geq 1$$

Matrix tensile failure $\left(\sigma_{22}+\sigma_{33}\geq 0\right)$
(3)$${\left(\frac{\sigma_{11}}{X_{T}}\right)}^{2}+{\left(\frac{\sigma_{12}}{S_{12}}\right)}^{2}+\frac{{\left(\sigma_{22}\right)}^{2}}{Y_{T}Y_{C}}+\frac{\sigma_{22}}{Y_{T}}+\frac{\sigma_{22}}{Y_{C}}\geq 1$$

Matrix failure by compression $\left(\sigma_{22}+\sigma_{33}<0\right)$
(4)$${\left(\frac{\sigma_{11}}{X_{T}}\right)}^{2}+{\left(\frac{\sigma_{12}}{S_{12}}\right)}^{2}+\frac{{\left(\sigma_{22}\right)}^{2}}{Y_{T}Y_{C}}+\frac{\sigma_{22}}{Y_{T}}+\frac{\sigma_{22}}{Y_{C}}\geq 1$$
where *X_T_*, *X_C_*, *Y_T_*, and *Y_C_* are the tensile and compressive strengths in the longitudinal and transverse directions, respectively, and $\sigma_{ij}\left(i,j=1,2,3\right)$ are the Corsi stress tensor components. *S*_12_ is the shear strength in the fiber and transverse directions, *S*_13_ is the shear strength in the fiber and thickness directions, and *S*_23_ is the shear strength in the transverse and thickness directions.

In this paper, the damage criterion of composite laminates was modeled using the user-defined subroutine (VUMAT) employed in Abaqus/Explicit to analyze the damage mechanism of composite laminates. Table 1 lists the strength parameters associated with the carbon fiber plies used in the finite element simulations, with parameters obtained from the manufacturer. It should be noted that the failure elements will be eliminated from the geometry in order to ensure stability during the analysis and will not be considered in the next calculation step.

The maximum tensile stress theory is used for the damage criterion. When the maximum tensile stress of the material reaches a certain limit value (i.e., the strength limit measured by the axial tensile test of the material), the material fractures; the strength formula is as follows:(5)$$\sigma_{1}\leq \left[\sigma \right]$$
where, $\sigma_{1}$ is the maximum tensile stress of the material and [σ] is the strength limit of the material.

#### 2.2.2. Rubber Intrinsic Structure Model and Parameters

Rubber is a viscoelastic solid and its viscoelasticity is a time-dependent function. Considering the short duration of the low-velocity impact process, the foamed silicone rubber is treated as a super-elastomer and reduced to the Cauchy elasticity problem. Considering only purely mechanical processes under isothermal conditions, the model is further simplified to an isotropic substance, so that the strain energy density function W is a function of the right deformation tensor [C] (or the left deformation tensor [B]), or expressed as a function of the elongation ratio so that the strain energy density function per unit volume can be expressed as:(6)$$W=W\left(I_{1},I_{2},J=I_{3}^{1/2}\right)\;\mathrm{or}\;W=W\left(\lambda_{1},\lambda_{2},\lambda_{3}\right)$$

The performance of foamed silicone rubber differs from that of conventional rubber in that, at the initial stage of loading of foamed silicone rubber, the rubber undergoes compressible deformation (Poisson’s ratio v ≈ 0) due to the presence of internal pores, and when the pores in the rubber are compacted, the material again exhibits properties similar to those of conventional rubber [20].

Using the porosity *f_0_* to represent the pore size of the foamed silicone rubber, the strain energy density W of the foamed silicone rubber can then be determined by the material parameters of the base rubber, i.e., the porosity *f_0_* at the initial time and the invariants *I*_1_, *I*_2_, *I*_3_ of the deformation tensor B.
(7)$$W=W\left(I_{1},I_{2},I_{3},f_{0}\right)$$
where $f_{0}=1-\rho_{P}/\rho_{S}$, $\rho_{P}$ is the density of foamed silicone rubber and $\rho_{S}$ is the density of solid rubber. According to the density of solid rubber range of 1100~1200 kg/m^3^, take $\rho_{S}=1150\mathrm{kg}/m^{3}$.

To obtain the expression for the strain energy density of foamed silicone rubber, the model of a thick-walled spherical shell can be used to represent the internal porous structure [21]. Assuming that the elongation ratios of the spherical shell in three directions are λ_1_, λ_2_, and λ_3_ for each point on the solid part, there is a corresponding strain energy density W and corresponding invariants *I*_1_, *I*_2_, and *I*_3_.

Although the foamed silicone rubber as a whole exhibits extremely high compressibility, the compressibility of the solid part of the rubber is small and can be considered incompressible. For incompressible materials, the Jacobi determinant *j* of the deformation gradient tensor F can be expressed as *j* = det(*F*) = 1, and since *B* = *FF*^T^, then *I*_3_ = det(*B*) = 1 holds at any point within the spherical shell. The strain energy density is simplified as W = W (*I*_1_, *I*_2_) and is related only to the first and second invariants *I*_1_ and *I*_2_. The invariants *I*_1_ and *I*_2_ at any point in the spherical shell can be expressed as a function of the macroscopic principal elongation ${\hat{\lambda }}_{1},{\hat{\lambda }}_{2},{\hat{\lambda }}_{3}$; the macroscopic invariants ${\hat{I}}_{1},{\hat{I}}_{2},\hat{j}$ (Jacobi determinant of the macroscopic deformation gradient); and the coordinates *X*, *Y*, and *Z* of the reference configuration, i.e.,
(8)$$I_{1}=\frac{1}{{\hat{j}}^{2/3}}\left[\hat{I_{1}}\psi^{2}+\frac{1}{R^{2}}\left({\hat{\lambda_{1}}}^{2}X^{2}+{\hat{\lambda_{2}}}^{2}Y^{2}+{\hat{\lambda_{3}}}^{2}Z^{2}\right)\left(\psi^{-4}-\psi^{2}\right)\right]$$
(9)$$I_{2}={\hat{j}}^{2/3}\left[\frac{\hat{I_{1}}}{{\hat{j}}^{2}\psi^{2}}+\frac{1}{R^{2}}\left(\frac{X^{2}}{{\hat{\lambda_{1}}}^{2}}+\frac{Y^{2}}{{\hat{\lambda_{2}}}^{2}}+\frac{Z^{2}}{{\hat{\lambda_{3}}}^{2}}\right)\left(\psi^{4}-\psi^{-2}\right)\right]$$
where the mapping function from the reference configuration to the current configuration is $\psi =\psi \left(R\right)={\left[1+\left(\hat{j}-1\right){\left(b/R\right)}^{3}\right]}^{1/3}$ and the radial distance from the reference point to the origin of the coordinates *R* = (*X*^2^ + *Y*^2^ + *Z*^2^)^1/2^. Using the above equation, the integration of the strain energy density over the entire spherical shell representative unit can be obtained and divided by the volume of the ball *V*_0_ = 4πb^3^/3, *b* being the outer radius of the ball. The average strain energy density of the foamed silicone rubber as a whole is obtained $\hat{W}$ as follows:(10)$$\hat{W}=\frac{1}{V_{0}}\int_{bf_{0}^{1/3}}^{b}\int_{0}^{2\pi }\int_{0}^{\pi }W\left(I_{1},I_{2},I_{3}\right)R^{2}\sin\theta d\theta d\varphi dR$$
where *θ*, *φ*, *R* are the three components in the spherical coordinate system of the reference configuration, and 0 ≤ *θ* ≤ π and 0 ≤ *φ* ≤ 2π are the standard spherical angles. From this, the average strain energy density can be used to obtain the average Corsi stress of the foamed silicone rubber as a whole.
(11)$$\hat{\sigma }=\frac{2}{\hat{j}}\frac{\partial \hat{W}}{\partial \hat{I_{1}}}\hat{B}+\frac{2}{\hat{j}}\frac{\partial W}{\partial \hat{I_{2}}}\left(\hat{I_{1}}\hat{B}-{\hat{B}}^{2}\right)+\frac{\partial \hat{W}}{\partial \hat{j}}I$$
where *I* is the second-order unit tensor.

If the solid rubber is treated as a Yeoh material, the derivative term of ${\partial /\partial I}_{2}$ is considered to be much smaller than ${\partial /\partial I}_{1}$, so the derivative term of *I*_2_ in the strain energy density can be discarded, and the expression of the strain energy density *W_Y_* is
(12)$$W_{Y}=\sum_{i=1}^{3}C_{i0}{\left(\lambda_{1}^{\alpha_{i}}+\lambda_{2}^{\alpha_{i}}+\lambda_{3}^{\alpha_{i}}-3\right)}^{i}+W_{v}$$
where *C*_10_, *C*_20_, *C*_30_, are material parameters and *Wv* is zero if the material is incompressible and no volume deformation occurs. According to the uniaxial tensile test data in the published literature, *C*_10_ = 0.6, *C*_20_ = −0.21, and *C*_30_ = 0.08.

Using the above equation to integrate over the interior of the spherical shell, the average strain energy density is thus obtained as
(13)$$\hat{W_{Y}}={\sum_{i=1}^{3}C_{i0}\left[\hat{I_{1}}\left[2-\frac{1}{\hat{j}}-\frac{f_{0}+2\left(\hat{j}-1\right)}{{\hat{j}}^{2/3}\eta^{1/3}}\right]-3\left(1-f_{0}\right)\right]}^{i}$$

#### 2.2.3. Finite Element Modeling

In the numerical simulation, the sandwich structure has a size of 100 mm × 100 mm × 8.9 mm, which accounts for a small proportion of the overall structure due to the small thickness of the coating and is considered as an isotropic material with a density of 400 kg/m^3^ and Young’s modulus of 6 GPa in this study species, using eight-node linear hexahedral cells, shrinkage integration, and hourglass control. The thickness of the unidirectional carbon fiber layer is 0.15 mm, the density is 300 g/m^2^, and the fiber plate uses a universal continuous shell grid within the eight-node quadrilateral face, reduction integration, hourglass control, and finite film strain (SC8R). The foamed silicone rubber thickness is 5 mm, the density is 950 kg/m^3^, and the rubber sandwich uses four-node linear tetrahedral cells (C3D4). A finite element model of the rubber sandwich structure is generated and analyzed using ABAQUS/Explicit software with fixed boundary conditions and 48 fully constrained supports, applying symmetric edge specimens on the upper and lower surfaces with displacements and rotation angles set to zero in the x, y, and z directions. In this paper, it is assumed that the pores are initially spherical and uniformly distributed, making the porous material initially isotropic. Figure 2 shows the finite element model of the sandwich structure under impact loading.

The impact angle of the falling hammer is varied in the numerical simulation, as shown in Figure 3. The impact angle *θ* is defined as the angle between the impact axis and the specimen. The two components of the vertical impact velocity are normal velocity *V_n_* and tangential velocity *V_t_*. In this paper, the impact velocities are set as 4.970 m/s, 5.495 m/s, 5.973 m/s, 6.419 m/s, 6.830 m/s, and 8 m/s, and the impact angles are 30°, 45°, 60°, 75°, and 90°, respectively.

### 2.3. Model Validation

The comparison between the experimental results and the finite element results when the stacking order of the rubber sandwich structure is W (woven fiber)_3_/Core/W (woven fiber)_3_, the porosity is *f*_0_ = 0.17, the impact velocity is 2.97 m/s, and the impact angle is 90° is shown in Figure 4. It can be seen from the figure that the curves have the same variation trend and the relative error between the finite element simulation and the experimental results is within 15%, indicating that the experimental results have good correlation with the simulation results. The reason for the error may be that the internal pores of rubber are assumed to be uniformly distributed, have uniform pore size, and have a certain number of spherical pores in the calculation process, while the actual internal pore structure of rubber is more complex.

## 3. Results and Discussion

### 3.1. Energy Change

In order to gain insight into the effect of the impact angle on the impact resistance of the sandwich structure, more than 30 impact simulations were conducted in this study, with the falling hammer impacting the sandwich structure at 4.970 m/s, 5.495 m/s, 5.973 m/s, 6.419 m/s, 6.830 m/s, and 8 m/s, respectively. *v*_r_ is the residual velocity. When *v*_r_ is negative, it means the falling hammer bounces off the sandwich structure, and a positive value means the falling hammer penetrates the sandwich structure.

The relationship between the initial and residual velocities of the impact at different angles was obtained from the data in Table 2, and the curve was fitted using the Levenberg–Marquardt optimization algorithm based on the data and the following equation (R^2^ > 0.99) as shown in Figure 5, with the expression valid only for *v_i_* > *v_threshold_*.
(14)$$v_{r}=A{\left(v_{{}_{i}}^{B}-v_{threshold}^{B}\right)}^{1/B}$$
where A, B, and *v_threshold_* are the fitting parameters; *v_i_* is the initial velocity of the impact; and *v_threshold_* is defined as the velocity penetration threshold at a given angle.

As shown in Figure 6, the penetration thresholds at 30°, 45°, 60°, 75°, and 90° can be obtained from the regression curves in Figure 5 as 6.747 m/s, 5.968 m/s, 5.640 m/s, 5.482 m/s, and 5.466 m/s, respectively. From this, it can be seen that the impact resistance decreases with the increase of impact angle by 11.5%, 16.4%, 18.7%, and 18.9% when the impact angle is greater than 45° and, with the increase of impact angle, the difference of impact threshold becomes smaller and smaller. When the impact angle is greater than 75°, compared with 90°, the difference of impact resistance performance is only 2.9%; at this time the impact angle has less impact resistance performance.

In summary, the smaller the impact angle is, the more energy is absorbed; therefore, 90° is the most unfavorable impact angle to the structure deformation, and avoiding the impact from the frontal side of the sandwich structure can effectively reduce the damage degree of the sandwich structure. Analysis: as the impact angle decreases, the separation between the core layer and the panel along the outer side of the falling hammer trajectory becomes more and more obvious, and the damage mode of the sandwich structure changes from shear damage to tensile damage and the path of the falling hammer through the sandwich structure becomes longer as the impact angle decreases, so the absorbed energy increases. This phenomenon indicates that the change of the path of the drop hammer through the sandwich structure at different impact angles has a greater effect on the energy absorption of the rubber sandwich structure.

### 3.2. Mechanical Response Analysis

A diagram comparing positive impact (impact angle of 90°) with oblique impact (impact angles of 30°, 45°, 60°, and 75°) is shown in Figure 7. Under the low-speed oblique impact condition, the hemispherical drop hammer first makes a point of contact with the sandwich structure and then continuously presses down on the sandwich structure. At this time, the contact area between the drop hammer and the specimen becomes elliptical. Because of the angle between the drop hammer and the sandwich structure, the drop hammer will produce a certain amount of slip on the upper panel of the sandwich structure, so the loading process of the drop hammer is tangential slip and normal loading at the same time, the deformation shape of the sandwich structure is constantly changing and expanding tangentially, and the position of the drop hammer on the sandwich structure is also constantly changing. When the kinetic energy of the hammer is completely dissipated, the elastic strain energy of the sandwich structure starts to be released and the hammer starts to rebound. In the rebound process of the drop hammer, because the drop hammer only has the translational freedom in the *Z*-axis direction, the drop hammer will still slide on the surface of the sandwich structure during the rebound process, and the frictional dissipation energy will be generated until the drop hammer is out of contact with the sandwich structure.

However, under positive impact conditions, the shape of the contact deformation generated during the loading of the drop hammer on the sandwich structure is always circular and always in a symmetrical state. Although there is a slight slip of the drop hammer on the sandwich structure when the deformation occurs, the slip is so small that the friction loss is almost negligible, so the kinetic energy of the drop hammer can be considered to be completely dissipated by the sandwich structure in the form of internal damage. When the kinetic energy of the falling hammer is completely dissipated, the elastic strain energy accumulated in the sandwich structure starts to be released, and the sandwich structure starts to push the falling hammer to rebound. Since there is no tilting relationship between the hammer and the sandwich structure at this time, there is almost no friction loss during the rebound process, so the process can be considered as only the conversion of the elastic strain energy of the sandwich structure to the kinetic energy of the hammer until the hammer is out of contact with the sandwich structure and the positive impact process is finished.

The contact force displacement curves of the sandwich structure under different impact angles with impact velocities of 4.970 m/s and 6.830 m/s are shown in Figure 8. It can be seen from the curves that when the impact velocity is 4.970 m/s, the load goes through two rises and two falls, and the sudden fall of the load is due to the compression damage of the substrate in the impact area. In the rebound phase, the curve is relatively stable, and finally, the hammer head is out of contact with the sandwich structure, and the load disappears. When the impact speed is 6.830 m/s, the sandwich structure is completely penetrated at various impact angles, and no rebound occurs. It can be seen from the figure that in the case of oblique impact, whether the falling hammer partially bounces or the falling hammer completely penetrates the core structure, the peak contact force is positively correlated with the impact angle, and the displacement under the maximum load is negatively correlated with the impact angle. In the positive impact case, the friction force is very small and almost negligible, and its maximum contact force is slightly less than 75°.

When the impact angle is less than 45°, the tangential impact force becomes larger than the normal impact force, and the slip of the hammer becomes easier. As the hammer slides, it changes its position on the sandwich structure. As the position changes, the units that were farther away from the hammer begin to join the impact resistance process of the hammer, and these newly added units will continuously deform in flexure to absorb the kinetic energy of the hammer. Although the impact angle increases when the hammer slide increases the new unit damage, the easier the slide also means that the greater the frictional energy dissipation, which leads to a large amount of kinetic energy of the hammer being dissipated through the slide in the form of frictional heat, so the actual impact energy loaded into the sandwich structure plate is not much, which leads to the structure only being the expansion of the damage area, while tilting the impact of the hammer and the contact area of the sandwich structure. At the same time, the contact area between the falling hammer and the sandwich structure is relatively increased, so the overall damage will be reduced.

### 3.3. Failure Modes

Different failure modes were observed at different velocities. This subsection is analyzed for *v* = 4.970 m/s (less than the penetration threshold) and 6.830 m/s (greater than the penetration threshold) velocities, respectively.

Figure 9 shows the final damage and stress clouds of each profile of the sandwich structure under different angles of impact at *v* = 4.970 m/s. According to Figure 9, it can be seen that the structure will always produce craters after the impact of the falling hammer regardless of the change of the impact angle, but the center of the craters will change, which is more obvious when the impact angle is small. As the impact angle decreases from 90° (positive impact), the slip becomes more and more, and the center of the crater moves in the direction of the slip, accordingly. The nature of the crater is the plastic deformation of the sandwich structure to absorb the energy after the impact load is applied. Comprehensive Table 2 and Figure 9 show that at *v* = 4.970 m/s, the impact head bounced to different degrees at various impact angles, and the sandwich structure was not completely penetrated, but the bottom panel had different degrees of damage, of which 90° damage was the most serious. When the impact hammer head penetrated the coating, the upper panel, and the core, the coating of the sandwich structure was broken, the upper surface had fiber fracture, and the core was broken, which were caused by the shearing process, and these damages appeared independent of the impact velocity. As the impact velocity was below the penetration threshold, the bottom panel showed matrix cracks. In addition, it is obvious that when excluding the energy consumed by slip friction, most of the impact energy loaded onto the structure is mainly consumed through plastic deformation of the upper panel and flexural deformation of the core unit, and the deformation of the lower panel under low-energy impact is extremely small and almost does not participate in energy absorption. When the impact is oblique, craters appear in the coating, the upper panel, and the core, and cracks extend from the edges of the craters.

Figure 10 shows the damage to the coating, the upper panel, the rubber core, and the bottom panel of the sandwich structure at *v* = 6.830 m/s and at different angles. Combining Table 2 and Figure 10, it can be seen that the sandwich structure was penetrated at *v* = 6.830 m/s at various impact angles. Compared with an impact velocity greater than the penetration threshold at *v* = 4.970 m/s, there is extensive fiber fracture in the bottom panel due to the indentation of the falling hammer, and further indentation expands the fiber fracture area and the spalling of the bottom panel, which leads to the cracking of the bottom panel matrix. The edges of the bottom panel fiber fracture were jagged, consistent with the edge morphology of tensile fracture, and it was determined that the bending tensile stress along the fiber direction caused the bottom panel fiber fracture. When the impact angle was 30°, the impact velocity was greater than the penetration threshold, the path of the falling hammer along the plane of the sandwich structure was too long, and the contact area was large, extensive fiber fracture and delamination occurred in the top panel, and the bottom panel showed a damage pattern similar to that of normal impact. Similarly, combined with Figure 9 and Figure 10, it can be seen that at the same impact energy, the oblique impact always produced less damage in the lower panel of the sandwich structure than the positive impact.

## 4. Conclusions

In this paper, the red-eared slider turtle is used as the prototype of the bionic design of the foamed silicone rubber sandwich structure, and five impact angles are selected by numerical simulation to study the effect of impact angle on the impact resistance of the rubber sandwich structure at different impact velocities. The response of the sandwich structure is evaluated by using the penetration threshold and the absorption energy respectively, and the damage mechanism of the sandwich structure is analyzed. The following conclusions were obtained:Numerical methods for calculating the structure of the foamed silicone rubber sandwich using a rubber intrinsic model with porosity and a three-dimensional Hashin criterion are effective;Based on the simulation data, the curve relationship between the initial velocity and the residual velocity was fitted using the Levenberg–Marquardt optimization algorithm, and the penetration thresholds for impact angles of 30°, 45°, 60°, 75°, and 90° were 6.747 m/s, 5.968 m/s, 5.640 m/s, and 5.482 m/s, and the impact resistance decreased by 11.5%, 16.4%, 18.7%, and 18.9% with the increase of impact angle;When the impact angle is greater than 45°, with the impact angle increases, the difference between the impact threshold is smaller and smaller. When the impact angle is greater than 75°, compared with 90°, the impact resistance difference is only 2.9%; at this time, the impact angle has less impact resistance performance;The impact angle has an obvious effect on the energy absorption characteristics of the rubber sandwich structure. At a certain impact speed, the smaller the impact angle, the longer the path of the falling hammer along the plane of the sandwich structure, the larger the contact area, the lighter the degree of damage, and the greater the energy absorbed by the sandwich structure; therefore, 90° is the most unfavorable impact angle for structural deformation, and avoiding the impact from the front of the sandwich structure can effectively reduce the degree of damage to the sandwich structure;The damage patterns of positive impact and oblique impact on the upper panel are different. For positive impact, the upper panel of the sandwich structure had fiber fracture caused by the shearing process. For oblique impact, fiber fracture and multiple cracks were produced at the edge of the falling hammer due to the larger contact area that the falling hammer passed through and then removed a large amount of material from the upper panel.

## Figures and Tables

**Figure 1 polymers-15-00688-f001:**
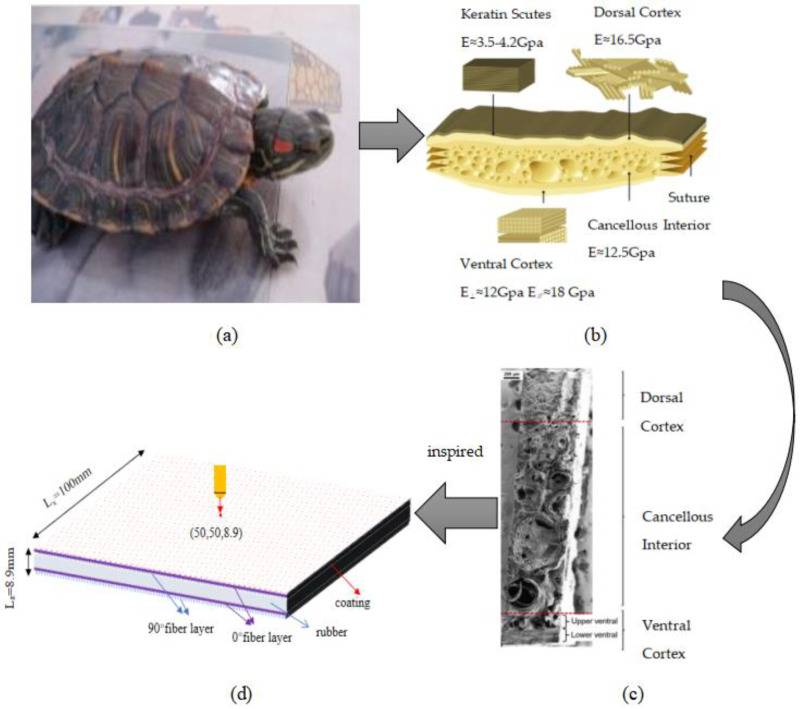
Bionic red-eared slider turtle sandwich structure design: (**a**) A macroscopic morphology of a turtle shell. (**b**) [6] A cross-sectional view of the turtle shell carapace showing composite layers. (**c**) [6] A μCT reconstruction modified using an SEM fractography showing the sagittal surface of the carapace rib. The fractography reveals the dorsal and ventral cortices and the cancellous interior of the rib. (**d**) Schematic diagram of stacking sequence of bionic sandwich structure and impact region.

**Figure 2 polymers-15-00688-f002:**
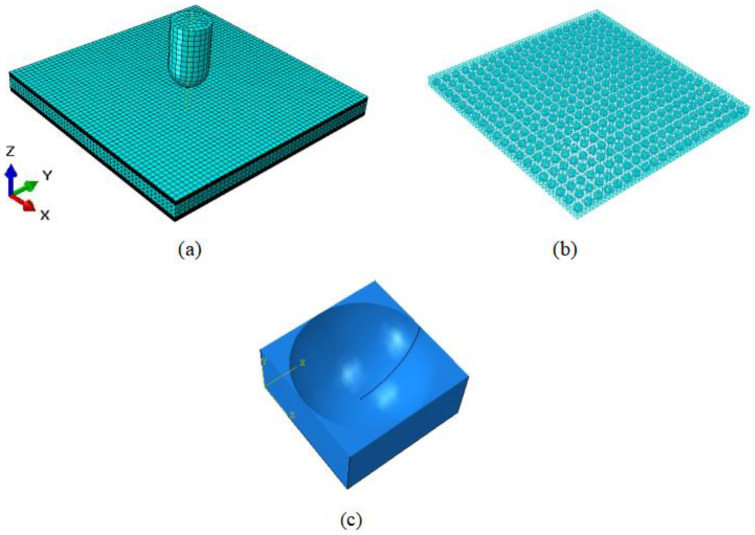
Finite element model of composite laminate under impact loading. (**a**) The whole model. (**b**) The rubber model: there are 400 same-sized spherical voids uniformly distributed in the model. (**c**) Half of void-containing cube.

**Figure 3 polymers-15-00688-f003:**
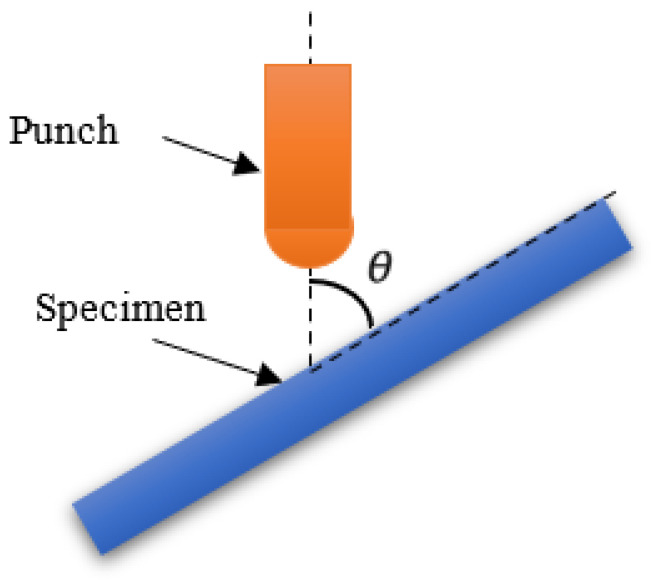
Schematic diagram of the inclined impact.

**Figure 4 polymers-15-00688-f004:**
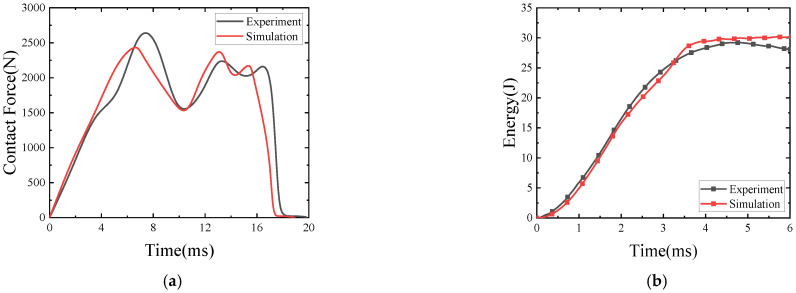
Comparison of experimental results and simulation results for sandwich structures with porosity *f*_0_ = 0.17 at 30 J impact energy, (**a**) Contact Force-deflection; (**b**) Energy-time.

**Figure 5 polymers-15-00688-f005:**
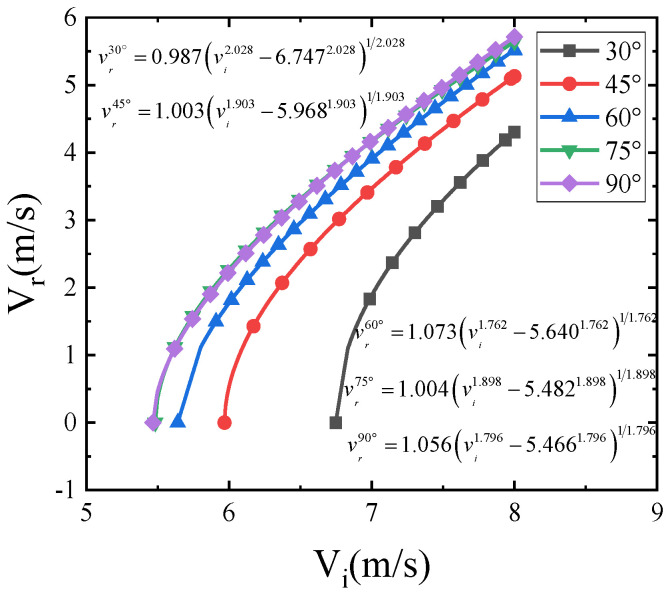
Residual velocity vs. impact velocity for different impact angles.

**Figure 6 polymers-15-00688-f006:**
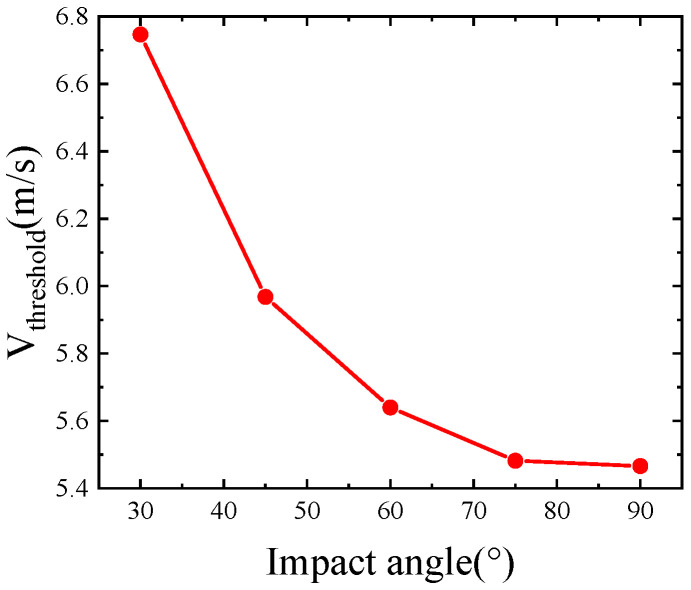
Impact thresholds at different impact angles.

**Figure 7 polymers-15-00688-f007:**
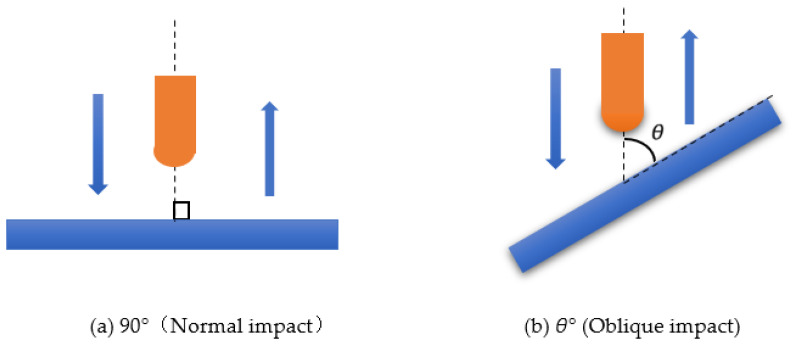
Comparison diagram of impact and rebound process of normal impact and oblique impact.

**Figure 8 polymers-15-00688-f008:**
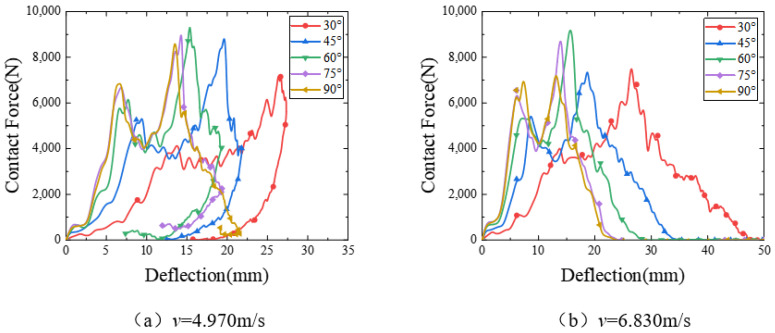
Contact force-deflection curves at different impact angles.

**Figure 9 polymers-15-00688-f009:**
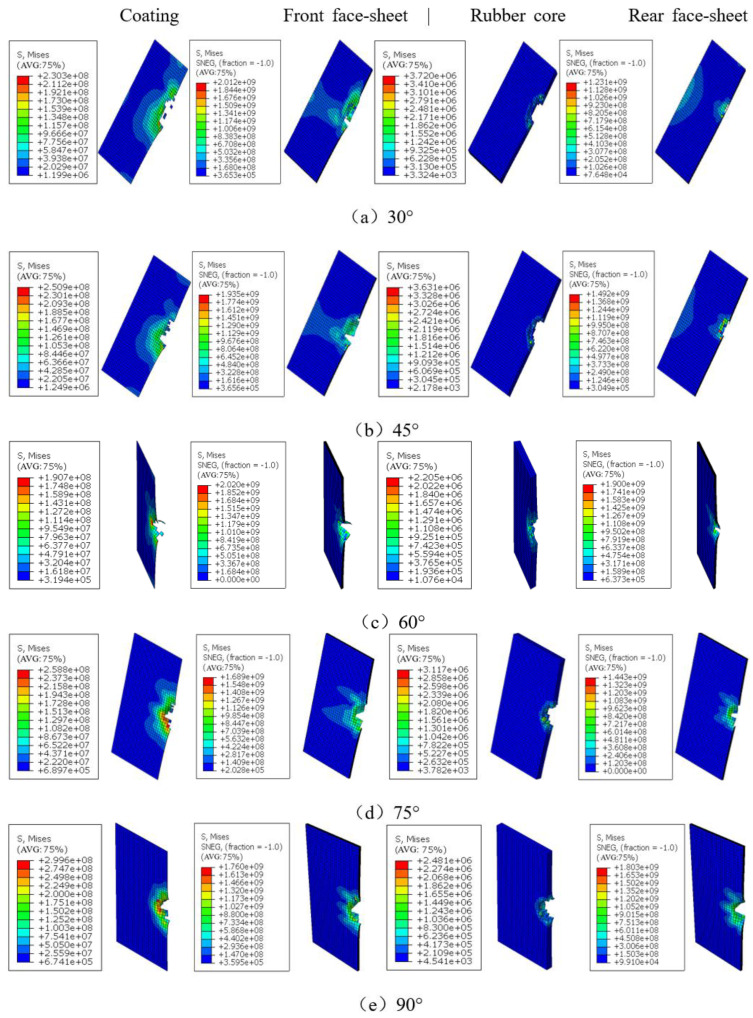
Damage of the coating, upper panel, core, and bottom panel of sandwich structure at different impact angles when *v* = 4.970 m/s.

**Figure 10 polymers-15-00688-f010:**
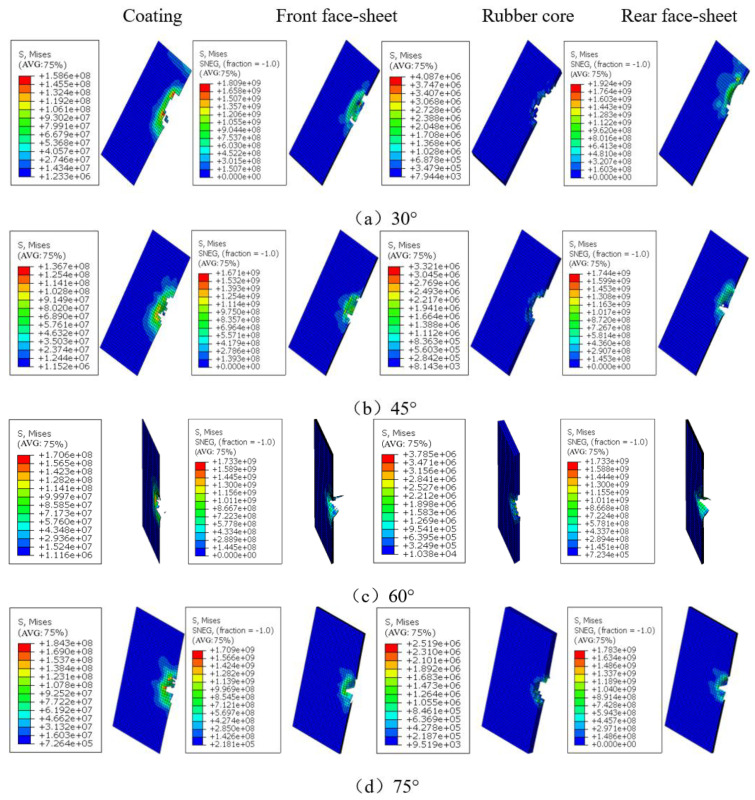
Damage of the coating, upper panel, core, and bottom panel of sandwich structure at different impact angles when *v* = 6.830 m/s.

**Table 1 polymers-15-00688-t001:** The strength parameters adopted in FE simulation.

Parameters	Symbol	Value	Units
Young’s modulus	*E*_11_, *E*_22_, *E*_33_	135, 8.8, 8.8	GPa
Poisson’s ratio	*v*_12_, *v*_13_, *v*_23_	0.33, 0.33, 0.35	_
Shear modulus	*G*_12_, *G*_13_, *G*_23_	4.47, 4.47, 4.0	GPa
Ultimate tensile stress	*X_T_*, *Y_T_*, *Z_T_*	1548, 55.8, 55.8	MPa
Ultimate compressive stress	*X_C_*, *Y_C_*, *Z_C_*	1226, 131, 131	MPa
Ultimate shear stress	*S*_12_, *S*_13_, *S*_23_	89.9, 89.9, 51.2	MPa

**Table 2 polymers-15-00688-t002:** Residual velocity and impact velocity for different impact angles.

*θ* *V*_i_ (m/s)	30 *v*_r_ (m/s)	45 *v*_r_ (m/s)	60 *v*_r_ (m/s)	75 *v*_r_ (m/s)	90 *v*_r_ (m/s)
4.970	−1.445	−1.365	−1.308	−0.921	−0.433
5.495	−0.854	−0.738	−0.153	0.321	0.431
5.973	−0.766	0.190	1.66411	2.185	2.170
6.419	−0.143	2.255	2.814	3.176	3.083
6.830	1.082	3.089	3.603	3.910	3.959
8.000	4.327	5.119	5.518	5.634	5.696

## Data Availability

The data that support the findings of this study are available upon reasonable request from the authors.

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
