# Peer review of "Effect of Impact Angle on the Impact Mechanical Properties of Bionic Foamed Silicone Rubber Sandwich Structure"

_polymers, 2023, doi:10.3390/polym15030688_

Round 1

Reviewer 1 Report

abstract has too many ands and long sentences should be revised.

l97 perhaps remove we

l111 figs b and c  not clear

l151 eq 6 confusing

l243 fig 4 needs to be consistent between experimental and fea, lines and points etc

l275 change we again

l325 mention about 300

l382 why mises stress for outer sheets, what are the units and what does this indicate

Author Response

Point 1: abstract has too many ands and long sentences should be revised.

Response 1: We appreciate and agree with your comments and have modified the long sentences in the abstract as appropriate.

Point 2: l97 perhaps remove we.

Response 2: We agree with this suggestion and have modified the sentence, as shown in l110.

Point 3: l111 figs b and c not clear.

Response 3: We agree with this suggestion and figs b and c have been enlarged, as shown in l121- l24.

Point 4: l151 eq 6 confusing.

Response 4: We agree with this suggestion and have modified the eq 6, as shown below l167.

Point 5: l243 fig 4 needs to be consistent between experimental and fea, lines and points etc.

Response 5: We appreciate your suggestion, because the actual pore structure inside the rubber is more complex, there are infinite pores inside the rubber, and in the numerical simulation calculation process due to the setting of calculation accuracy, according to the porosity inside the rubber assuming uniform pore size, only a certain number of spherical pores can appear. This differs from the test to a certain extent, leading to a certain error between the numerical simulation and the test. However, the overall trend of numerical simulation and test in this paper is consistent with a maximum error of 10%, within 15%, which can prove the accuracy of numerical simulation to a certain extent. In the next work, we will continue to precise the conditions and optimize the computational resources to reduce the magnitude of the errors.

Point 6: l275 change we again.

Response 6: We agree with this suggestion and have modified the sentence, as shown in l301.

Point 7: l325 mention about 300.

Response 7: I'm really sorry we didn't find the word 30° in l325 and the paragraphs before and after, this issue has not been modified.

Point 8: l382 why mises stress for outer sheets, what are the units and what does this indicate.

Response 8: In the simulation of this paper, the unit of Mises stress is Mpa. The Mises stress of each layer can visually reflect the stress magnitude and failure of each place, where the stress is more likely to be damaged and more likely to fail earlier under the same condition. The Mises stress cloud at different impact angles can also visualize the effect of impact angle on each layer of the sandwich structure.

Reviewer 2 Report

The manuscript entitled "  Effect of impact angle on the impact mechanical properties of bionic foamed silicone rubber sandwich structure” by  Di Zhang et al. describes, a red-eared slider turtle is used as a prototype for the bionic design of the foamed silicone rubber sandwich structure, and the effect of impact angle on the performance of the foamed silicone rubber sandwich structure against low-velocity impact is studied by the finite element method.

This is an interesting contribution in the field of Bio-Inspired Materials, however in my opinion before its publication in the Polymer journal the authors need to address comments that I have regarding their manuscript.

Below please find my comments:

Introduction: The introduction part could be rewritten. The literature part presents a amount of general information without focusing on information important from the point of view of the conducted research and its location and justification against others background of Bio-Inspired Materials.

Please provide a better description of the state of the art in this interesting topic, including the most important findings of other recently published works using Bio-Inspired Materials.

Page 2 Line 96: Please give more information about the concept the armor is composed of a sandwich composite structure. Reference the used the FEA models, also include images if possible.

Page 3, line 109-110: The manuscript needs to improve the quality of the figures 1. Please carefully check. (Include SEM. Porosity images).

Page 3-4 Line 116-139: Please explain with scientific arguments the use of this damage criterion.

Page 7, line 217: The manuscript needs to improve the quality of the figures 2. Please carefully check

After the authors work out these observations, their paper can be accepted for publication.

Author Response

Point 1: Introduction: The introduction part could be rewritten. The literature part presents a amount of general information without focusing on information important from the point of view of the conducted research and its location and justification against others background of Bio-Inspired Materials. Please provide a better description of the state of the art in this interesting topic, including the most important findings of other recently published works using Bio-Inspired Materials.

Response 1:  We appreciate and agree with your comments and have read the literature on the subject and added appropriate references to the writing of this article to make the subject more obvious.

Point 2: Page 2 Line 96: Please give more information about the concept the armor is composed of a sandwich composite structure. Reference the used the FEA models, also include images if possible.

Response 2: We agree with this suggestion and add a μCT reconstruction modified using a SEM fractograph showing the saggital surface of the carapace rib. The fractograph reveals the dorsal and ventral cortices and the cancellous interior of the rib. As can be seen from the figures 1(c), the armor consists of a sandwich composite structure, as shown in Page 3, line 124.

Point 3: Page 3, line 109-110: The manuscript needs to improve the quality of the figures 1. Please carefully check. (Include SEM. Porosity images).

Response 3: We agree with this suggestion and have improved the quality of the figures 1, as shown in Page 3, line 121-124.

Point 4: Page 3-4 Line 116-139: Please explain with scientific arguments the use of this damage criterion.

Response 4: Composite materials have been widely used because of their excellent mechanical properties. The strength analysis is an inevitable problem in the application process. The finite element analysis adopts the progressive loss method. According to the progressive damage theory, the material will not be destroyed immediately after the damage occurs, but the material property will degrade, and the units around the damaged unit will bear more load. With the aggravation of damage, the carrying capacity of the material will continuously decline or the damaged units will gradually increase until the overall damage. The commonly used degradation models of material properties include instantaneous unloading model and gradual degradation model. Instantaneous degradation model directly reduces material properties such as stiffness coefficient and elastic modulus constant to a certain low level and remains unchanged. The gradual degradation model holds that the material properties of the unit will not suddenly decline at the initial damage stage, but gradually reduce the material properties with the aggravation of damage until the unit exceeds the specified value and completely fails. The progressive degradation model based on energy method is adopted in this paper. The common three-dimensional Hashin criteria are used to determine the tensile and compressive failure of the fiber. When either failure mode in lines 116 to 139 takes effect, Abaqus immediately removes the destructive element until the entire fiber fails.

Point 5: Page 7, line 217: The manuscript needs to improve the quality of the figures 2. Please carefully check.

Response 5: We agree with this suggestion and have improved the quality of the figures 2, and we added two pictures to clarify the internal porosity of rubber, as shown in Page 7, line 232-238.

Round 2

Reviewer 2 Report

The revisions made by the authors have effectively addressed all concerns and have greatly improved the manuscript. I recommend that it be accepted for publication.